# Nanoparticle-Based Therapeutic Strategies for Enhanced Pancreatic Ductal Adenocarcinoma Immunotherapy

**DOI:** 10.3390/pharmaceutics14102033

**Published:** 2022-09-24

**Authors:** Wanting Hou, Biao Yang, Hong Zhu

**Affiliations:** 1Department of Medical Oncology Cancer Center, West China Hospital, Sichuan University, Chengdu 610017, China; 2Department of Gastroenterology, West China Hospital, Sichuan University, Chengdu 610017, China

**Keywords:** pancreatic ductal adenocarcinoma, nanomedicine, drug delivery, immunotherapy

## Abstract

Immunotherapy has dramatically changed prognosis for patients with malignant tumors. However, as a non-immunogenic tumor, pancreatic ductal adenocarcinoma (PDAC) has a low response to immunotherapy. Factors that contribute to the inefficiency of PDAC immunotherapy include the tumor microenvironment (TME) and its dense stroma, which acts as a barrier for drug delivery and immune cell infiltration. Recent studies have shown that nanoparticle-based therapeutic strategies have more promising applications in improving drug delivery and reversing the immunosuppressive TME for PDAC. Therefore, nanomaterial-based therapeutic approaches are expected to enhance the effectiveness of immunotherapy and improve prognosis of patients with PDAC. Here, we outline the status and dilemma of PDAC immunotherapy, and summarize the latest advances in nanoparticle-based treatment strategies to enhance the efficacy of PDAC immunotherapy.

## 1. Introduction

Pancreatic cancer is a lethal tumor with an increasing incidence. With the advancements in tumor diagnosis and therapy, several malignant tumor-associated mortality rates have decreased; however, the pancreatic-cancer-associated mortality rate remains high. The reasons behind the high mortality rates for pancreatic cancer include difficulty in achieving early diagnosis and lack of effective treatment options [1]. Pancreatic ductal adenocarcinoma (PDAC) is the main type of pathological pancreatic cancer, accounting over 90% of all pancreatic cancers. Moreover, it has the worst response to treatment and highest mortality rate of all pathological types of pancreatic cancer [2].

Radical resection is a relatively effective treatment option for patients with PDAC. However, the lack of specific clinical symptoms leads to late diagnosis and dismal prognosis. Over 50% of patients presenting with liver or peritoneal metastases are unresectable at the time of diagnosis [3]. Even for patients who have their tumor lesions removed, recurrence is difficult to avoid due to the presence of microscopic lesions. Hence, following surgery, patients require adjuvant treatment. The current therapies have a limited effect on the survival of patients with PDAC [4]. The main treatment methods include chemotherapy, radiotherapy, molecular targeted therapy, and immunotherapy. Of these, chemotherapy is the most effective. Standard chemotherapy regimens for PDAC present modest survival benefits, with resistance ultimately unavoidable [1]. Even for patients with advanced and distant metastatic PDAC, who have received the standard chemotherapy, the five-year survival rate is still less than 5% [4]. Thus, developing more effective treatment options is important for improving the prognosis and survival of patients with PDAC.

Immunotherapy, especially immune checkpoint inhibitors (ICIs), has shown remarkable efficacy in many malignant tumors; however, the current immunotherapy response is dismal in patients with PDAC [5]. The reason for this is complex. High desmoplasia plays an important role in the poor immunotherapy efficacy for PDAC. The PDAC stroma accounts for over 80% of tumor mass. The tumor comprises various cells, including cancer-associated fibroblasts (CAFs), pancreatic stellate cells (PSCs), tumor cells, immune cells, and endothelial cells, that secrete scores of cytokines and induce the formation of the extracellular matrix (ECM). This dense stroma acts as a “physical barrier” to obstruct the antitumor agent delivery and as an immunosuppressive biological barrier to limit immune active cell infiltration [6]. Stromal cells can also induce immunosuppressive tumor cells [7]. Moreover, the infiltration of immunosuppressive cells, including regulatory T (Treg) cells, tumor-associated macrophages (TAMs), and myeloid-derived suppressor cells (MDSCs), was reportedly rich, while immunosupportive cells, such as intratumoral CD8^+^ T cells, were relatively scarce in the PDAC tumor microenvironment [8,9]. Additionally, PDAC is a hypovascular tumor accompanied by hypoxia and a high interstitial fluid pressure (IFP) in the tumor microenvironment (TME), which impairs drug delivery and leads to immunosuppression [10].

The current immunotherapeutic approaches have shown a limited improvement in the prognosis for patients with PDAC [11]. However, several studies reported that a positive response to the therapy and improved prognosis in patients with PDAC are associated with immune activation [12,13,14,15,16]. Therefore, immunotherapy remains a promising treatment option for patients with PDAC. New therapeutic strategies for improving effectiveness of immunotherapy for PDAC include combining existing strategies and exploring new therapeutic targets [17,18,19]. Multiple molecules (including non-coding RNAs) have been identified as the potential research targets to enhance the immunotherapy efficacy for PDAC [20,21]. However, delivery of these small molecules or co-delivery of antitumor agents to the tumor’s interior has become an important obstacle to overcome. Additionally, improved approaches are required to enable the drugs to cross the dense stroma of PDAC to reach the deeper part of the tumor. Recently, nanoparticles have shown promise as carriers for antitumor agent delivery. Complex molecules, such as non-coding RNAs (e.g., siRNA and microRNA), can also be delivered using nanoparticles as the carriers [22,23]. The advantages of these nanoparticles include controllable size and drug release, prolonged circulation, reduced side effects, co-delivery of multiple therapeutic agents, and site-specific drug delivery [23]. Two nano-formulations have been approved by the U.S. Food and Drug Administration (FDA) for PDAC therapy. The first is an albumin-bound paclitaxel called nab-paclitaxel. Compared to the traditional paclitaxel, nab-paclitaxel shows a higher plasma clearance, better distribution, lower toxicity, and better therapeutic results [24]. Based on results from a phase 3 MPACT trial, the FDA approved nab-paclitaxel as a delivery agent for the chemotherapy drug gemcitabine as the first-line treatment for metastatic PDAC [25]. Another nano-formulation approved by the FDA for PDAC treatment is the liposomal formulation of irinotecan. Liposomal irinotecan plus 5-fluorouracil (5-FU) and leucovorin was approved by the FDA and used as second-line treatment option for metastatic patients with PDAC who were resistant to gemcitabine. Compared to free irinotecan, the liposomal formulations of irinotecan presented lower toxicity and better therapeutic results [26].

Several preclinical studies have presented data on the benefits of nanoparticle-based therapeutic strategies for PDAC immunotherapy. Nanoparticles can remodel the tumor stroma, improve drug delivery efficiency, regulate the immunosuppressive microenvironment, activate antitumor immunity, and induce a stronger antitumor effect [27,28]. In this review, we summarize the current status of PDAC immunotherapy and discuss obstacles in PDAC research. Additionally, we outline the potential role of nanoparticle-based therapeutic strategies in improving drug delivery and regulating immunosuppressive TME of PDAC.

## 2. Status and Dilemma of Immunotherapy for PDAC

Tumor immunotherapy involves killing tumor cells by activating the immune response of the body. The current immunotherapy approaches include cancer vaccines, ICIs, adoptive cell transfer (ACT), monoclonal antibodies, and cytokine-mediated therapies [29]. Of these therapeutic modalities, ICIs are currently the only immunotherapy strategy approved for clinical use in multiple solid tumors (e.g., non-small cell lung cancer, melanoma, renal cell cancer, and colorectal cancer) [30]. The main targets of the ICIs include cytotoxic T-lymphocyte-associated protein 4 (CTLA-4), programmed cell death protein-1 (PD-1), and programmed cell death ligand-1 (PD-L1). Monoclonal antibodies, including anti-PD-L1 inhibitors (e.g., atezolizumab, durvalumab, and avelumab), anti-PD-1 inhibitors (e.g., pembrolizumab and nivolumab), and CTLA4 receptor inhibitors (e.g., ipilimumab and tremelimumab), have been approved by the FDA for several solid tumor treatments [30,31]. ICI monoclonal antibodies, as single drugs or combination strategies, were studied in PDAC clinical trials [32,33,34,35,36,37,38,39,40,41,42,43]. Other immunotherapeutic modalities (e.g., cancer vaccines, ACT, and cytokine-mediated therapies) were also studied in pancreatic cancer clinical trials; however, their clinical application remains to be demonstrated [44,45,46]. This section summarizes the key clinical trial results for PDAC immunotherapy (especially ICI and ICI combined with chemotherapy/radiotherapy) and outlines the influence of the tumor immune microenvironment (TIME) on the efficacy of PDAC immunotherapy.

### 2.1. Status of Immunotherapy for PDAC

Reported clinical trials on the immunotherapy strategies for PDAC include ICIs, cancer vaccines, adoptive cellular immunotherapy (ACI), and cytokine-mediated therapies. The results for key clinical trials on PDAC immunotherapy are summarized in Table 1 and Appendix A.

Studies on ICIs and their use in a combined strategy account for the majority of the PDAC clinical research. ICIs release the inhibitory brakes in T cells, which strongly activates the immune system and induces antitumor immune reactions [30,31]. The first reported ICI in a PDAC clinical trial was ipilimumab, a CTLA-4 inhibitor. In this phase 2 clinical trial, 27 patients with locally advanced or metastatic PDAC were treated with the ipilimumab for two cycles. Although one case with a delayed reaction was reported, no objective response was reported in this study [32]. In 2012, the safety and activity of anti-PD-L1 antibody BMS-936559 were evaluated in patients with advanced cancer. In this multicenter phase 1 trial, 14 patients with PDAC were recruited and treated with the BMS-936559. Similar to the ipilimumab trial result, no objective response was observed in patients with PDAC after the anti-PD-L1 inhibitor treatment [33]. These results indicate that PDAC does not respond well to ICI monotherapy. By 2020, KEYNOTE-158 showed that patients with solid tumors with high microsatellite instability (MSI-H)/mismatch repair-deficient (dMMR) can benefit from anti-PD-L1 monotherapy. Here, the overall response rate (ORR) of 22 patients with PDAC with MSI-H was 18.2%. This led the FDA to approve pembrolizumab for treatment in patients with solid tumors with MSI-H/dMMR, including PDAC [34]. However, it is worth noting that the ORR for PDAC was much lower than that of other tumors in the study, and only a very small proportion of patients with PDAC (below 2%) had MSI-H [35]. Efficacy of the dual ICI combination therapy was also poor in patients with PDAC. In a phase II clinical study, 65 metastatic patients with PDAC received combination therapy with the anti-PD-L1 antibody durvalumab and anti-CTLA-4 antibody tremelimumab. This combination treatment did not improve the prognosis of patients with PDAC. Moreover, its adverse effects increased significantly [36].

Following the failure of single- and double-ICI combination therapies, researchers started exploring whether combination therapy strategies could improve the efficacy of PDAC immunotherapy. It was suggested that chemotherapy agents may boost the immune response and potentially overcome PDAC resistance to ICIs [37]. Therefore, a combination therapy strategy of chemotherapy with ICI was attempted in patients with PDAC. Gemcitabine combined with anti-CTLA-4 antibodies (i.e., tremelimumab and ipilimumab) was evaluated in two phase I trials. In one clinical trial, tremelimumab combined with gemcitabine was used to treat metastatic PDAC. The median overall survival (OS) was 7.4 months. At the end of the treatment, 2 of the 34 patients achieved partial response (PR) [38]. In the second trial, ipilimumab was combined with gemcitabine for the treatment of advanced PDAC. The first reported response (PR and stable disease (SD)) was at 43% [39]. The final result from this study was reported in 2020 with an ORR of 14%. The median progression-free survival (PFS) and median OS were 2.78 and 6.90 months, respectively [19]. Gemcitabine combined with anti-PD-1 and anti-PD-L1 inhibitors has also been administered to patients with advanced PDAC. A phase Ib/II trial reported on the use of the anti-PD-1 inhibitor pembrolizumab, in combination with gemcitabine and nab-paclitaxel, for treatment of advanced solid tumors. Among the PDAC arms, disease control rates (PR and SD) were at 100%. The median PFS and OS were 9.1 and 15.0 months, respectively [40]. In another phase I study, the anti-PD-1 inhibitor nivolumab was combined with nab-paclitaxel and gemcitabine as treatment for patients with advanced pancreatic cancer. A response rate was observed in 18% of the patients. The median PFS and OS were 5.5 and 9.9 months, respectively [41]. Recently, a randomized phase III trial tested a new PD-1 antibody called sintilimab, in combination with modified FOLFIRINOX versus FOLFIRINOX alone as the first- or second-line therapy for metastatic and recurrent PDAC. In this study, an improved ORR (50% vs. 23.9%) was observed in the sintilimab and modified FOLFIRINOX combination groups, but OS and PFS did not improve [42]. Additionally, the results of a phase II study showed that there was no significant difference in the OS and PFS between the dual ICI and chemotherapy combined groups and chemotherapy alone [43]. These results suggest that, although the combination of chemotherapy and immunotherapy slightly improved the response rate in patients with PDAC relative to immunotherapy alone, there was still no significant improvement in their OS.

Radiation therapy presents another promising strategy for enhancing the therapeutic effects of PDAC immunotherapy. Radiation therapy may increase the response to immunotherapy through an abscopal effect [47]. In a preclinical study, radiation promoted the release of tumor-specific antigens and enhanced priming of tumor-specific T cells in PDAC mouse models [48]. However, the current clinical trial data show that the combination of radiotherapy and ICI only has moderate therapeutic benefits for patients with PDAC, and does not improve their survival [49,50,51].

Other monotherapies and combination therapies were widely explored in PDAC clinical trials (e.g., tumor vaccines, oncolytic viruses, ACI, and monoclonal antibodies; Appendix A); however, these trials only reported slight benefits in the response rates in patients with PDAC. Overall, the existing immunotherapy strategies did not significantly improve survival of patients with PDAC, and were limited by the small number of participants; hence, these therapies need to be tested within a larger subject population.

**Table 1 pharmaceutics-14-02033-t001:** Published clinical trials of ICIs in PDAC.

	Therapeutic Strategy	Phase	Stage	Number of Patients	Objective Response Rate	Median PFS (Months)	Median OS (Months)	Immune-Related Adverse Events (≥Grade 3)	Publication Year	NCT Number	Ref.
Sigle ICI	ipilimumab (anti-CTLA-4)	II	Pre-treated LAPC/mPDAC	27	0	NA	NA	11.1% (3/27)	2010	NCT00112580	[32]
	BMS-936559 (anti-PD-L1)	I	advanced PDAC	14	0	NA	NA	NA	2012	NCT00729664	[33]
	pembrolizumab (anti-PD-1)	II	advanced PDAC	22	18.2 (all dMMR patients)	2.1	4	NA	2020	NCT01876511	[34]
Double ICIs	durvalumab (anti-PD-L1) + tremelimumab (anti-CTLA-4) vs. durvalumab	II	mPDAC	32 vs. 33	3.1% vs. 0%	1.5 vs. 1.5	3.1 vs. 3.6	22% vs. 6%	2019	NCT02558894	[36]
ICIs + chemotherapy	tremelimumab (anti-CTLA-4) + GEM	I	chemotherapy-naive mPDAC	34	2 PR	NA	7.4	NA	2014	NCT00556023	[38]
	ipilimumab (anti-CTLA-4) + GEM	Ib	advanced PDAC	16	43% PR + SD	2.5	8.5	NA	2016	NCT01473940	[39]
	ipilimumab (anti-CTLA-4) + GEM	Ib	advanced PDAC	21	14%	2.78	6.9	19%	2020	NCT01473940	[19]
	GEM + Nab-P + pembrolizumab (anti-PD-1)	Ib/II	mPDAC	17	100% PR + SD	9.1	15	53%	2018	NCT02331251	[40]
	nivolumab (anti-PD-1) + GEM + Nab-P	I	advanced PDAC	50	9%	5.5	9.9	96%	2020	NCT02309177	[41]
	sintilimab (anti-PD-1) + FOLFIRINOX vs. FOLFIRINOX	III	metastatic and recurrent PDAC	55 vs. 55	50% vs. 23.9%	5.9 vs. 5.73	10.9 vs. 10.8	5.7% (sintilimab + FOLFIRINOX)	2022	NCT03977272	[42]
	GEM + Nab-P + durvalumab (anti-PD-L1) + tremelimumab (anti-CTLA-4) vs. GEM + Nab-P	II	mPDAC	119 vs. 61	30.3% vs. 23.0%	5.5 vs. 5.4	9.8 vs. 8.8	38% vs. 20%	2020	NCT02879318	[43]
ICIs + radiotherapy	SBRT + nivolumab (anti-PD-1) vs. SBRT + nivolumab (anti-PD-1) + ipilimumab (anti-CTLA-4)	II	Refractory mPDAC	41 vs. 43	clinical benefit rate 17.1% vs. 37.2%	1.7 vs. 1.6	3.8 vs. 3.8	24.4% vs. 30.2%	2022	NCT02866383	[49]
	radiation + nivolumab (anti-PD-1) + ipilimumab (anti-CTLA-4)	II	metastatic MSS PDAC	25	18%	2.7	6.1	56%	2021	NCT03104439	[50]
	durvalumab (anti-PD-L1) + SBRT 8 Gy vs. durvalumab (anti-PD-L1) + SBRT 25 Gy vs. durvalumab (anti-PD-L1)/tremelimumab(anti-CTLA-4) + SBRT 8 Gy vs. durvalumab (anti-PD-L1)/tremelimumab(anti-CTLA-4) + SBRT 25 Gy	I	mPDAC	14 vs. 10 vs. 19 vs. 16	5.1%	1.7 vs. 2.5 vs. 0.9 vs. 2.3	3.3 vs. 9.0 vs. 2.1 vs. 4.2	7.1% vs. 33.3% vs. 21.1% vs. 62.5%	2020	NCT02311361	[51]

GEM: gemcitabine; Nab-P: nab-paclitaxel; LAPC: locally advanced pancreatic cancer; mPDAC: metastatic pancreatic ductal adenocarcinoma; SBRT: stereotactic body radiation therapy; NA: not available.

### 2.2. Dilemma in Immunotherapy for PDAC

The difficulty in immunotherapy for PDAC is mainly due to its unique TIME (Figure 1). Tumor, immune, and stromal cells, and extracellular components constitute the TIME. The TIME is crucial for the induction and maintenance of malignant tumor phenotypes. Moreover, it can directly determine the patient prognosis and response to immunotherapy [52]. According to the infiltration of immune cells in the TIME, tumors can be classified as “hot” (immunologically active) and “cold” (immunologically inactive). “Hot tumors” are characterized by an abundant immune infiltration and a better response to immunotherapy. PDAC is generally considered a “cold tumor” with poor immunogenicity [53]. Abundant immunosuppressive cells exist in the TIME of the PDAC, whereas immunosupportive cells (e.g., CD8^+^ T cells) are limited. The immunosuppressive cells in the TIME of PDAC include MDSCs, TAMs, and Tregs. These cells secrete cytokines to sustain the immunosuppressive TIME in PDAC cells. Moreover, PDAC has a low mutational burden, resulting in low neoantigen levels and a reduced ability in T cells to recognize tumors [54]. Therefore, successful induction and activation of effector T cells in the TIME are required for an effective PDAC antitumor immune response [55].

PDAC is characterized by its abundant desmoplastic stroma that contains a variety of cellular components. These components include immune cells, PSCs, CAFs, and endothelial cells, with CAFs representing the main cells in the PDAC stroma [6]. However, the origin and functional characteristics of CAFs remains unclear. PSCs are generally considered the main source of CAFs [56]. Heterogeneous CAFs play a variety of roles in the modeling of TIME for the PDAC. For example, myofibroblastic CAFs can induce a dense fibrous tissue formation to create a barrier that protects cancer cells from being recognized by the immune system. Inflammatory CAFs can produce cytokines that induce TAM differentiation and recruit immunosuppressive cells, such as Tregs and MDSCs [57]. Together, the cellular and ECM form an immunosuppressive biological barrier, which acts as a drug delivery barrier. Breaking this barrier would vastly improve drug delivery, and consequently, improve the immune response of PDAC. However, the tumor stroma is an important barrier that limits the tumor metastasis [58]. An indiscriminate targeting of the stroma may promote PDAC invasion [59]. Thus, rational stromal remodeling would be an ideal research direction for improving PDAC immunotherapy.

Additionally, PDAC is characterized as a hypovascular tumor. The hypovascularity and poor perfusion of PDAC not only limits the delivery of antitumor agents, but also induces hypoxia in the TIME [10]. Hypoxia is another important obstacle for PDAC immunotherapy. Hypoxia can induce and maintain an immunosuppressive TME and activate PSCs to mediate fibrosis [60]. Therapeutic strategies targeting optimized hypoxia have shown synergistic improvements in immunotherapy in the preclinical studies for pancreatic cancer [61,62]. In a phase I clinical trial, evofosfamide (prodrug that alleviates hypoxia) combined with ipilimumab was tested in advanced solid malignancies. Here, seven patients with pancreatic cancer were enrolled, and two achieved SD [63]. Further clinical trials are required to confirm this result. Tumor vascular normalization is an emerging strategy for alleviating hypoxia and enhancing cancer immunotherapy [64]. Some studies proposed strategies to normalize tumor vasculature in pancreatic cancer, and their potential role in PDAC immunotherapy was proven in preclinical models [65,66]. These results are yet to be demonstrated in clinical trials.

Overall, the specific immunosuppressive TIME, high levels of desmoplasia, hypovascularity, and extreme hypoxia are typical pathological features of PDAC, and are some of the main reasons for its low response to immunotherapy. A more comprehensive investigation of the PDAC TIME is needed to find the effective immunotherapy targets.

## 3. Nanoparticle-Based Therapeutic Strategies for Enhanced PDAC Immunotherapy

Two key problems hinder the therapeutic effects of PDAC immunotherapy: the inadequate response caused by the immunosuppressive TIME and the delivery obstacle presented by the dense tumor stroma and hypovascularization [8]. Recent studies show that nanoparticles could help solve these problems. Nanoparticles can not only be applied to regulate the immune system, but, more importantly, can enable, modulate, and penetrate the dense stroma of PDAC [23,27]. For example, nano-albumin-bound (Nab)-paclitaxel, a nano-based drug commonly used in PDAC treatment, was proven to deplete the tumor stroma through the interaction between the albumin and secreted protein acid, and is rich in cysteine [67]. Additionally, the size of the nanoparticles is adjustable. Small-sized nanoparticles can pass through the tumor stroma where molecular drugs cannot penetrate, which allows nanoparticles to co-deliver stroma depletion and anti-tumor/immunomodulatory agents [68]. Recently, the use of nano drug delivery devices coupled with stroma depletion has emerged as a promising treatment for PDAC [69]. Furthermore, several studies have tested nanoparticle-based photothermal therapy (PTT), photodynamic therapy (PDT), chemodynamic therapy (CDT), and sonodynamic therapy (SDT) for treatment of PDAC, and the results were promising in the preclinical models [70,71,72,73,74,75,76,77,78]. Thus, nanoparticle-based therapeutic strategies have provided new avenues for the PDAC immunotherapy. Here, we summarize the reported nanoparticle-based therapeutic strategies for the PDAC immunotherapy (Table 2) and discuss their prospects and challenges.

### 3.1. Nanoparticle-Based Immunogenic Cell Death Strategies for Enhanced PDAC Immunotherapy

Immunogenic cell death (ICD) is a specific type of cancer cell death. When the tumor cells are stimulated by external stimuli, the dying tumor cells release damage-associated molecular patterns (DAMPs) that activate tumor-specific immune responses and boost antitumor responses. Additionally, ICD induces the continuous release of tumor antigens, which can help turn cold tumors into hot tumors. Moreover, ICD leads to the activation and recruitment of cytotoxic T cell lymphocytes (CTLs) [108]. Therefore, ICDs could work synergistically with ICIs to increase the patient immune response rates. Some chemotherapeutic drugs, oncolytic viruses, physicochemical therapies, photodynamic therapies, and radiotherapies can induce ICD [108]. Nanomedicines can reportedly enhance the ICD-inducible agents to exert stronger immune effects, owing to their proven advantages in drug delivery. This is especially suitable for solid tumors with poor drug delivery, such as PDAC [109].

Oxaliplatin is a commonly used chemotherapeutic agent for PDAC treatment and is an ICD-inducible agent [110]. Compared to free oxaliplatin treatment, pancreatic tumor cells treated with nanoparticles encapsulating oxaliplatin release more DAMPs and induce stronger dendritic cell immune responses and a higher percentage of tumor-infiltrating activated cytotoxic T lymphocytes [79]. Additionally, nanoparticle capsules can help reduce the oxaliplatin toxicity. Liu et al. developed a mesoporous silica nanoparticle (MSNP)-based platform for the high-dose loading of platinum chemotherapeutic agents. Using an orthotopic Kras-derived PDAC model, they proved that MSNP not only improved the intratumoral administration and pharmacokinetics of the oxaliplatin, but also maintained stability of the colloid after intravenous injection and reduced its toxicity [80]. Nanoparticles can also realize an effective transportation of molecular entities with antitumor and ICD effects. Shen et al. proposed a sequential receptor-mediated mixed-charge targeted drug delivery system to administer effective components from the Euphorbia plant for the treatment of pancreatic cancer. Here, the modified nanomicelles induced ICD and penetrated the tumor vessel walls [81]. Therefore, nanoparticle-based therapy strategies could maximize the role of the ICD drugs, enhance their immune enhancement and antitumor effects, and have a potential application value in PDAC therapy.

### 3.2. Nanoparticle-Based Immune Modulation Strategies for Enhanced PDAC Immunotherapy

Regulation of the immune microenvironment is key to improving the effectiveness of PDAC immunotherapy. The TIME of the PDAC is frequently characterized by a low number of tumor-infiltrating lymphocytes (TILs) and a high number of Tregs, M2 TAMs, and MDSCs. The latter are related to the poor immune response for PDAC. Repolarization of the TAM phenotype from M2 to M1 is reportedly a promising strategy for PDAC treatment [111]. miR-125b can affect TAM repolarization. Parayath et al. conjugated the M2 peptides to hyaluronic acid-polyethyleneglycol (HA-PEG)/HA-polyethylenimine (HA-PEI) polymers to form self-assembled nanoparticles with miR-125b. This delivery system has a dual functionality: the HA polymer can target CD44 receptors on the surface of macrophages, and the M2 peptide can specifically target TAMs. The synthesized nanoparticles can effectively reprogram the M2 TAMs to the antitumoral M1 phenotype [82]. Moreover, PI3K-γ and colony-stimulating factor-1 (CSF-1) or CSF-1 receptor (CSF-1R) pathways are also involved in the infiltration and polarization of the M2 TAMs in pancreatic cancer [112]. Li et al. developed nanomicelles that co-deliver a PI3K-γ inhibitor and CSF-1R-siRNA. They proved that this nanomicelle not only reduced the M2 TAMs level and increased the M1 TAMs level, but also inhibited the tumor infiltration of MDSCs and effectively modulated the TIME of PDAC [83]. Cao et al. developed a nanoplatform that can co-deliver monoacylglycerol lipase siRNA (suppresses free fatty acid generation) and CB-2 siRNA (regulates macrophage phenotype). This nanoplatform can inhibit production of free fatty acids in the PDAC tumor cells and cut off nutritional supply to the tumor. Additionally, it can repolarize the TAMs into the tumor-inhibiting M1-like phenotype, reverse the immunosuppressive microenvironment, and produce synergistic antitumor effects [84]. MDSCs are another promising target for PDAC treatment [113]. However, therapeutic abrogation of MDSCs usually causes a compensatory recruitment; therefore, it cannot effectively improve the immunosuppressive TIME of PDAC [114]. Reportedly, inhibiting MDSC recruitment may weaken the immunosuppression [115]. Based on low-molecular-weight heparin (LMWH) inhibiting MDSC recruitment, Lu et al. developed LMWH-based nanoparticles loaded with paclitaxel. These nanoparticles not only improved the pancreatic TIME, but also inhibited spontaneous metastasis of the tumors [85]. In addition to the immunosuppressive cells, dysfunction and immunosuppressive antigen-presenting cells (APCs) are abundant in the immunosuppressive TIME of PDAC, which is also related to its low response to immunotherapy. Lorkowski et al. developed an immunostimulatory nanoparticle (immuno-NP), and co-loaded two immune agonists. The first agonist was a stimulator for the interferon pathway, and the second was an agonist for the Toll-like receptor 4 pathway. The combined introduction of these two agonists proved that immuno-NP can trigger robust activation and expansion of the APCs, and strengthen the immune response in PDAC [86].

Targeting immunomodulatory cytokines can also regulate the TIME. Hence, nanoparticle delivery of these cytokines can potentially achieve a more effective immunomodulation. Miao and Shen separately designed nanoparticles that co-load chemokine C-X-C motif ligand 12 (CXCL12) traps with Interleukin-10 (IL-10) or PD-L1 traps to synergistically change the immunosuppressive microenvironment of PDAC and modify the immunosuppressive TIME to allow the host’s immune system to kill the tumor cells [87,88]. Moreover, retinoic acid-inducible gene I (RIG-I)-like receptors (RLRs) induce secretion of IFN-α/β and other pro-inflammatory cytokines [116]. Das et al. developed an NP-mediated delivery for 5′-triphosphate double-stranded RNA, which is specific to Bcl2 and can bind to RLRs. These therapeutic nanoparticles modulated the TIME and inhibited tumor growth in the PDAC model [89]. Systemic administration of immunomodulatory drugs rarely reaches the tumor-draining lymph nodes (TDLNs); however, this is overcome when using the nanoparticles because they effectively deliver drugs to deep tumors and TDLNs. Han et al. combined Interleukin-12 (IL-12) microspheres with stereotactic body radiotherapy (SBRT) to achieve repolarization of the TIME for pancreatic cancer. This biodegradable microsphere not only enables local continuous administration, but can also deeply penetrate the TDLN of PDAC, thus achieving complete tumor inhibition [90].

Using nanoparticles as carriers can protect physically or chemically unstable molecules from exerting immunotherapeutic effects. A previous study found that the RNAi-based drugs have a high sequence specificity to target molecules and are considered more effective than antibodies for blocking immune checkpoints [117]. However, RNA is easily attacked by RNase in circulation. Jung et al. proposed the use of poly-based siRNA nanoparticles to target PD-L1. They showed that the modified PD-L1 siRNA nanodrug had better targeting than the traditional PD-L1 antibodies. This siPD-L1 nanoparticle not only showed an effective knockout of PD-L1 in the cancer cells, but also showed a strong antitumor immunity in a PDAC preclinical model [91]. Recently, microcapsules for oxygen delivery were proposed. Here, the oxygen microcapsules provided a stable oxygen delivery deep into the tumor, reversed hypoxia in the TME, and improved the performance of ICB in PDAC [62].

Nanoparticles can also be used as immune adjuvants in vaccines. Dong et al. used aluminum hydroxide nanoparticles together with PEI as an immune adjuvant for dendritic cell (DC) vaccines. This nano-adjuvant could enhance antigen transport and cross-presentation of DCs in a pancreatic cancer model, thus further improving the DC vaccine efficacy [92].

### 3.3. Nanoparticle-Based Chemoimmunotherapy for Enhanced PDAC Immunotherapy

Chemotherapy and immunotherapy can synergistically exert antitumor effects. Some chemotherapy drugs can induce ICD, which can then be enhanced in combination with ICIs or other immunomodulators. Therefore, the chemoimmunotherapy combination is rational [118]. However, during clinical studies, chemotherapy combined with ICI only presented a modest efficacy and did not improve prognosis for patients with PDAC [37,38,39,40,41,42,43]. This may be related to the inherently strong immunosuppressive TIME, lack of new antigens with low ICI response, or the dense stroma of PDAC. Nanoparticles can enhance the ICD of chemotherapeutic drugs, be used as immunomodulators, and act as carriers for co-delivery of immunomodulators and chemotherapeutic agents to achieve effective drug delivery to deep layers of the tumors [27,28]. Thus, nanoparticle-based chemoimmunotherapy presents a promising approach for enhancing the efficacy of PDAC immunotherapy.

Lu et al. first used MSNP to co-deliver oxaliplatin and an indoleamine 2,3-dioxygenase (IDO) inhibitor to treat PDAC. IDO is an enzyme that induces immunosuppressive activity. Therefore, IDO inhibitors can reverse the immunosuppressive TME. Additionally, oxaliplatin can induce ICD. Nanocarriers can improve the drug delivery and, consequently, the intratumoral concentration of IDO inhibitors and oxaliplatin. This synergistic approach boosted the antitumor immune response in a PDAC mouse model [93]. Based on a similar combination, Huang et al. developed cationic lipid-assisted polymeric nanoparticles to deliver the IDO1 siRNA. Using colorectal and pancreatic cancer mouse models, they proved that simultaneous administration of oxaliplatin and siIDO1 nanoparticles could achieve synergistic antitumor effects by promoting maturation of DCs, increasing TILs, and reducing the number of Tregs, which could then prevent tumor recurrence by provoking long-term antitumor immune memory [94].

Gemcitabine is a chemotherapeutic drug commonly used for PDAC treatment. Although gemcitabine cannot induce ICD [119], when modified with nanoparticles, it can be employed as a tumor-penetrating nanocarrier. In a previous study, a redox-responsive gemcitabine-conjugated polymer was used as the nanocarrier to co-load an IDO1 inhibitor (NLG919) and paclitaxel. The co-loaded micelles deeply penetrated the pancreatic tumor to induce immune-active antitumor activity in PDAC models [95]. Tong et al. developed a tumor-pH-sensitive polymer whose cavity was a hydrophobic gemcitabine prodrug. Nanopolymers can respond to the pH of the TME and transform into small particles to promote the in-depth delivery of gemcitabine. Additionally, these nanoparticles can modulate the TME, upregulate PD-L1 expression levels in tumor cells, and act synergistically with anti-PD-1 therapy [96]. Chen et al. designed TME-activatable charge-conversional micelles to co-load two prodrugs, GEM-C18 and NI-HJC0152. Once the lowered pH of the outer layer of PDAC is sensed, the micelle surface charge changes from negative to positive, releasing the prodrugs across the rich stroma of the PDAC. Additionally, NI-HJC0152 responds to hypoxia and releases HJC0152, inhibiting signal transducer and activator of transcription 3 (STAT3) and inducing immune activation. Thus, micelles play a dual role by reversing both the immunosuppressive TME and drug resistance in PDAC [97]. In addition to the pH response, special nanoparticle modifications can achieve effective delivery of antitumor drugs deep into the tumor. Hu et al. developed a negatively charged HA, which can achieve graded nanostructures and effectively deliver celastrol (a pentacyclic triterpenoid extracted from traditional Chinese medicine) and 1-methyltryptophan (IDO inhibitor) to a deep tumor site of the pancreas. The combination of celastrol with an IDO inhibitor, administered using nanoparticles, showed significantly enhanced tumor inhibition and downregulation of the immunosuppressive TIME [98]. Liposomal irinotecan is a nano-formulation currently approved for PDAC treatment. Liu et al. encapsulated irinotecan in a silicasome instead of a liposome and proved that it could induce a more powerful ICD response. Additionally, when combined with anti-PD-1 treatment, it led to a more pronounced survival improvement compared to anti-PD-1 combination therapy with free or liposomal irinotecan. Moreover, silicasome irinotecan is less leaky and toxic than liposomal irinotecan [99].

Targeting TAMs combined with chemotherapeutic drugs is another promising strategy for PDAC chemoimmunotherapy. Zhou et al. developed an exosome-based dual delivery biosystem for the galectin-9 siRNA and oxaliplatin prodrugs. Disruption of the galectin-9/dectin1 axis could reverse the PDAC immunosuppression of M2-TAMs, whereas oxaliplatin could trigger ICD. Thus, use of this biological material not only significantly increases accumulation of the drugs at the tumor site, but also enhances antitumor immunity in PDAC [22]. Similarly, Wang et al. developed a biomimetic dual-targeting nanomedicine based on gemcitabine and M2pep. Nano-formation enables the simultaneous targeted delivery of gemcitabine to the pancreatic tumor sites and TAMs repolarization to potentiate its therapeutic effects. This nano-formation also synergistically enhances the antitumor effect of PD-L1 antibodies in the PDAC [100].

These studies suggest that nanoparticles may not only serve as vehicles to improve the permeability and immunogenicity of existing chemotherapeutic agents, but, more importantly, may provide a flexible carrier for multiple chemoimmunotherapy combination options. Nanoparticle-based chemoimmunotherapy offers more possibilities for enhancing the efficacy of immunotherapy for PDAC.

### 3.4. Nanoparticle-Based Stroma Modulation for Enhanced PDAC Immunotherapy

The dense stroma in PDAC is a major obstacle for immunotherapy. It hinders the delivery of immunotherapeutic agents and infiltration of effector T lymphocytes. However, the dense stroma in PDAC is associated with the formation of an immunosuppressive TME (e.g., stromal cells like CAFs and PSCs), which can induce and recruit immunosuppressive lymphocytes as well as secrete immunosuppressive cytokines [120]. The ECM is the main non-cellular component of the PDAC stroma and is an important obstacle that hinders an effective drug delivery and effector T lymphocyte infiltration. The stroma cells and ECM are the main targets for PDAC tumor stroma regulation. Currently, stromal modulation combined with antitumor agents (e.g., chemotherapeutic or immunotherapeutic drugs) is regarded as an effective treatment option for PDAC management [121]. Several nanoparticle-based stoma modulation therapies have also been tested to enhance the PDAC immunotherapy.

Collagen and fibroblasts constitute the main components of the ECM and are the main targets for PDAC stroma modulation. Huang et al. proposed a nano-formation that co-loads an antifibrosis phosphate-modified α-mangostin (MP) and a plasmid encoding the immune-enhancing cytokine LIGHT (tumor necrosis factor superfamily 14, TNFSF14, CD258). In this nano-formation, called nanosapper, MP reversed the activation of CAFs, decreased collagen deposition, and relieved compressed vessels, while LIGHT normalized the tumor vessels, stimulated secretion of cytokines, recruited lymphocytes, and inhibited Tregs. Furthermore, this nanosapper can remodel TME and synergistically provide therapy with anti-PD-1 in PDAC models [101]. An acidic tumor extracellular pH-responsive clustered nanoparticle was developed to co-deliver a transforming growth factor-beta (TGF-β) receptor inhibitor and PD-L1 siRNA. The TGF-β receptor inhibitors can effectively inhibit activation of PSCs, reduce production of type I collagen, and remodel dense ECM. The siPD-L1 can silence the expression of PD-L1 in tumor cells. Therefore, this nano-formation can synergistically inhibit the growth of tumors in the PDAC model by activating the antitumor immune response [102]. The sonic hedgehog (SHH) signaling pathway plays an important role in the PDAC ECM synthesis. SHH signaling can activate the CAFs to produce a highly fibrotic stroma [122] and has been proposed as a promising target for the PDAC stroma remodeling [123]. However, SHH deletion reportedly caused tumors to become more aggressive in a PDAC mouse model [59]. Zhao et al. developed a polymeric micelle-based nano-formulation to co-deliver an SHH inhibitor and paclitaxel. This nano-formulation can increase the intratumoral vasculature density and promote CD8^+^ T cell infiltration through SHH inhibition, while paclitaxel can restrain tumor cell proliferation. They further proved that this nano-formulation could enhance the antitumor effect of anti-PD-1 monotherapy in a PDAC mouse model [103].

HA is another key component of the PDAC stroma and plays a role in increasing IFP, compressing tumor vessels, causing insufficient perfusion in the tumor, and hindering the delivery of antitumor agents [124]. PEGPH20 is a PEGylated recombinant human hyaluronidase that can reduce HA in the PDAC stroma [125]. PEGPH20 combined with chemotherapeutic agents showed promising antitumor effects in a PDAC preclinical model [126]. Blair et al. proposed a therapeutic strategy that combines PEGPH20 with GVAX (a granulocyte-macrophage colony-stimulating factor (GM-CSF) gene-transfected tumor cell vaccine), and proved that the single agent GVAX upregulates myeloid C-X-C chemokine receptor 4 (CXCR4) expression, while combined PEGPH20 modifying HA can decrease the CXCL12/CXCR4/chemokine receptor 7 (CCR7) immunosuppressive signaling axis, thus regulating myeloid cells, increasing T cell infiltration, and inducing memory effector T cells in PDAC. This combination can boost the antitumor immune response of GVAX [104]. Recently, researchers proposed a PEGPH20 combination with a focal adhesion kinase (FAK) inhibitor and an anti-PD-1 antibody to treat metastatic PDAC. In this combination strategy, PEGPH20 targets extracellular stromal HA, whereas FAK can regulate intracellular signaling in stromal and myeloid cells. The results proved that this dual stromal targeting combination can increase T cell infiltration and alter their phenotype towards effector memory T cells, thus sensitizing PDAC to anti-PD-1 therapy [105].

However, in clinical trials, PEGPH20 with chemotherapy did not improve the OS of patients with PDAC, and the incidence of adverse events was significantly higher in the combination therapy group [127,128]. Some researchers have proposed that the failure in PEGPH20 is related to the stroma, that it has protective effects in restraining PDAC tumor growth and progression, and that the stromal depletion may potentiate their metastatic capacity [129]. Thus, selective stroma modulation is the direction to enhance the efficacy of PDAC immunotherapy, instead of stroma deletion without discrimination. Chen et al. designed a nanoparticle co-loaded with paclitaxel and phosphorylated gemcitabine that selectively disrupted the central stroma and preserved the external stroma in a PDAC model. Thus, absence of the central stroma can effectively induce effector T lymphocyte infiltration, whereas the remaining external stroma can effectively prevent tumor metastasis. Additionally, these nanoparticles can modulate the immunosuppressive TIME of PDAC by augmenting the number of CTLs and restraining the percentage of Tregs [106]. Moreover, cancer–stroma interaction is an important factor in maintaining PDAC malignancy and affecting the PDAC response to immunotherapy [57]. Xie et al. designed cholesterol-modified polymeric CXCR4 antagonist nanoparticles for co-delivery of anti-miR-210 and siKRASG12D. This nano-formation has a triple-action for inhibiting PDAC: CXCR4 antagonists can block cancer–stroma interactions, anti-miR-210 can modulate the PDAC stroma through PSC inactivation, and siKRASG12D can kill pancreatic cancer cells. Since nanomedicines achieve a tumor-targeted delivery, mainly through the enhanced permeability and retention (EPR) effect following intravenous administration, this EPR effect is severely compromised in PDAC. Thus, they used an intraperitoneal administration to improve delivery and effective tumor penetration of nanoparticles. Combined therapy displayed an improved therapeutic effect in a PDAC model [107].

### 3.5. Nanoparticle-Based Photothermal/Photodynamic/Chemodynamic/Sonodynamic Therapy for Enhanced PDAC Immunotherapy

PTT converts light energy into heat energy to eliminate tumors [130]. PTT can induce ICD and elicit an immune response against cancer. Thus, PTT combined with immunotherapy presents a promising treatment for tumors. However, the dense stroma of PDAC impedes both laser penetration and the release of immune drugs deep into tumor tissue [130]. Yu et al. designed size-adjustable nanoparticles loaded with an immune checkpoint blocker, called BMS-202, to synergize with PTT. In this nanoplatform, the size-adjustable nanoparticles responded to fibroblast activation protein-α (specifically expressed on the fibrotic matrix) and a near-infrared (NIR) laser releasing small-sized nanoparticles. Small-sized nanoparticles loaded with BMS-202 were widely distributed within the tumor, and the immunosuppressive TME was alleviated. Meanwhile, the NIR laser-induced mild hyperthermia, reduced tumor hypoxia, and increased endogenous immune cell recruitment. Therefore, combination therapy inhibited not only tumor growth, but also tumor metastasis in the PDAC model [70]. Li et al. used a local tumor NIR photothermal immunotherapy to overcome the dense stroma of PDAC. They proposed supramolecular nanofibrils to co-load the clinically approved immunomodulatory thymopentin and photosensitizer indocyanine green for tumor photothermal immunotherapy. Using an orthotopic pancreatic tumor model, they showed that a local injection of the nanofibrils combined with NIR laser irradiation can effectively terminate tumor tissues while protecting normal tissues. Moreover, they demonstrated that this combination can promote proliferation and differentiation of antitumor immune cells, thus boosting tumor immunotherapy, while simultaneously eliminating tumor metastasis [71]. Sun et al. developed photosensitive dopamine-coated nanoparticles co-loaded with gemcitabine and NLG919 (IDO inhibitor) to construct nanoparticles for early- and late-stage metastatic cancer immunochemo-photothermal therapy. Using a PANC02 tumor-bearing mouse model, they found that the combination of the nanoparticles and laser irradiation enhanced the tumor inhibition effect. However, the abscopal effect of PTT combined with nanoparticles occurred mainly in late-stage tumor metastasis (large distal tumors), whereas in early-stage tumor metastasis (small distal tumors), no abscopal effect was observed [72]. Recently, Wang et al. proposed an interventional PTT (IPTT)-synergized immunotherapy guided by magnetic resonance imaging (MRI) for pancreatic cancer. They used amorphous iron oxide nanoparticles (IONs) loaded with photothermal agent’s indocyanine green (ICG) and immunoadjuvant imiquimod (IMQ) to construct a nanoplatform. This unique nanoplatform has multiple functions. First, it can serve as a contrast agent for the MRI, whereby an ION-assisted MRI is used to guide implementation of IPTT and monitor temperature distribution within the tumor and surrounding tissues during treatment, thus minimizing damage to surrounding healthy tissues. Second, it could act as a drug delivery carrier for IMQ and ICG, triggering a strong antitumor immunity. Third, it can act as a catalyst for the TME immune activation. Additionally, appropriately sized nanoparticles can penetrate the dense stroma in pancreatic cancer. Thus, IPTT-induced immune activation is synergistically enhanced by IMQ, while the IONs modulate the suppressive TME, thereby amplifying the antitumor immune effects. Together, they trigger a powerful antitumor systemic immunity to terminate both in situ tumors and metastasis [73].

PDT is another type of phototherapy that utilizes a specific wavelength to irradiate the tumor site, activate a photosensitive drug, and/or generate reactive oxygen species (ROS) to terminate the tumor. PDT was effective and promising in clinical trials for malignant tumors [131,132]. In phase I and I/II clinical trials, the safety of EUS-guided and verteporfin-PDT for locally advanced pancreatic cancer was confirmed [133,134]. Additionally, PDT can induce ICD and, consequently, be used in synergistic immunotherapy. However, there are still some obstacles in the use of PDT synergistic immunotherapy for PDAC. For example, PDT induces ICD while driving oxygen consumption and microvascular damage, further exacerbating hypoxia and glycolysis, which leads to lactate accumulation and aggravated immunosuppression in the TME. To solve this problem, Sun et al. designed HA-based supramolecular nanoparticles for co-delivery of photosensitizers, bromodomains, and extraterminal protein 4 inhibitors (BRD4i). In this nanosystem, BRD4i antagonizes the oncogene c-Myc and activate CTL, while HA contributes to long-term retention and deep tumor penetration for nanoparticles. They proved that this nanosystem combined with PDT could overcome the PDT-induced glycolysis and provoke a durable antitumor immunity in a pancreatic model [74]. Jang et al. used a tumor-derived reassembled exosome (R-Exo) as a delivery vehicle loaded with the photosensitizer chlorin e6 (Ce6-R-Exo). These modified exosomes retain their membrane proteins and have an average size, which allows for targeting of tumor and immune cells. The Ce6-R-Exo efficiently generated ROS-inducing cytokines that released under laser irradiation. Furthermore, Ce6-R-Exo can be captured by innate immune cells as antigens to boost immune responses, which can induce a long-term immune response and inhibit tumor growth, recurrence, and metastasis. Additionally, Ce6-R-Exo can be visualized through photoacoustic imaging. Therefore, Ce6-R-Exo provides a new strategy for an effective combination of PDT and immunotherapy [75].

SDT uses a low-intensity ultrasound that activates sonosensitizers to generate ROS and kill the tumor cells. Ultrasound penetrates deeper in the tissue than a laser; thus, SDT is more suitable for deep tumor treatment than the PDT/PTT [135]. Intratumoral oxygen is an important factor affecting SDT. The PDAC is an inherently hypoxic tumor. Hence, Nicholas et al. prepared pH-sensitive polymethacrylate-coated CaO_2_ nanoparticles, which could alleviate the tumor hypoxia. Here, Rose Bengal functioned as a sonosensitizer. A potent abscopal effect was observed in the SDT-treated PDAC animal models [76]. SDT can also induce ICD, thereby boosting immunotherapy. However, considering the dense stroma and poor blood perfusion in PDAC, the effectiveness of SDT may be limited. Chen et al. designed cavitation-assisted endoplasmic reticulum (ER)-targeted sonodynamic nanodroplets, which can achieve deep penetration as aided by ultrasound. Under the ultrasound irradiation, the modified sonosensitizer accumulates in the ER to generate large amounts of ROS in situ, which can further boost the efficacy of SDT. It was also demonstrated that the nanodroplets combined with SDT can enhance the effect of anti-PD-L1 immunotherapy in PDAC orthotopic and distant tumor models [77].

In addition to PDT/SDT, CDT is another ROS-mediated treatment method. However, unlike PDT/SDT, CDT causes damage to the tumor cells by converting ROS into highly toxic hydroxyl radicals (OH) through the Fenton reaction. Transition metal ions (i.e., Fe, Co, Ni, Cu, and Mn) are commonly used as reagents for CDT [136]. Chen et al. developed a tailored nanocomplex through the self-assembly of synthetic 4-(phosphonooxy) phenyl-2,4-dinitrobenzenesulfonate and Fe^3+^, followed by an HA modification to realize CDT for PDAC. By controlling the release of its components in a GSH-sensitive manner under the unique redox homeostasis of cancer cells and TAMs, the nanocomplex selectively triggers the Fenton reaction to induce oxidative damage in cancer cells, while repolarizing TAMs and deactivating stromal cells, thus attenuating stroma deposition. They further demonstrated that the CDT treatment provided a better tumor suppression than the conventional gemcitabine treatment, and did not show significant side effects in a PDAC mouse model; thus, CDT presents a promising strategy for PDAC [78].

In summary, PTT/PDT/SDT/CDT are promising strategies for cancer treatment because of their negligible toxicity and non-invasiveness. Additionally, these treatments can induce ICD, and thus can be used in combination with tumor immunotherapy. However, due to the insufficient penetration ability and special TME, traditional PTT/PDT/SDT/CDT has extremely limited effects in PDAC treatment. The application of nanoparticles facilitates use of these therapies for PDAC treatment. Nanoparticles can be designed for a deeper penetration or specific environmental responses, further exerting long-lasting antitumor effects. Furthermore, nanomedicine strategies could be designed to systematically co-deliver photosensitizer/sonosensitizer CDT agents and immunomodulators, thus boosting the immune-enhancing antitumor effects of these therapeutic strategies. These studies show that nanoparticle-based PDT/CDT/SDT/PTT has promising benefits in the PDAC immunotherapy and can be an important research direction for PDAC.

### 3.6. Nanoparticle-Based Imaging and Theranostic Agents for PDAC Immunotherapy

Nanoparticles also shows promise as imaging and theranostic agents [137]. For example, labelling of human mesenchymal stem cells with gold-poly-L-lysine nanocomplexes can achieve computed tomography (CT) in vivo tracking [138]. Iron oxide nanoparticles (IONs) radiolabeled with 68 Ga and 177 Lu could be used as potential agents for positron emission tomography (PET) for diagnosis and cancer therapy [139]. Chelator-free copper-64 incorporated with IONs showed good in vivo stability and could be used as a contrast agent for PET/MRI [140].

Several studies reported that modified multifunctional nanoparticles could be used as probes for imaging and theranostic agents in PDAC [141,142]. In this part, we focus on the value and potential applications of nanoparticles in imaging and theranostic roles in PDAC immunotherapy. The designed nanoparticles can increase efficiency of detection for PDAC, especially detection of early PDAC. For example, Wang et al. constructed Enolase 1 targeted superparamagnetic iron oxide nanoparticles that can increase MRI efficiency for detecting PDAC and facilitate early and accurate detection of PDAC [143]. Further, nanoparticles could be engineered to deliver imaging agents alone or in combination with anti-tumor agents, and thus used for imaging as well as image-guided and targeted cancer therapy. For example, Zhou et al. developed insulin-like growth factor 1 receptor receptor (IGF1R) multifunctional theranostic nanoparticles conjugated with recombinant human insulin-like growth factor 1 (IGF1) to magnetic IONs, carrying doxorubicin. These nanoparticles not only exhibited excellent anti-tumor effects in the orthotopic patient-derived xenograft (PDX) model, but could also be used as contrasts for MRI detection [144]. Image-guided treatment strategies can provide detailed information about each patient under treatment, which can then help monitor prognosis and adjust therapeutic interventions. Additionally, nanoparticles can be designed as probes to screen patients with PDAC suitable for immunotherapy. Liu et al. proposed nanoprobes that can sense tumor acidosis and hypoxic TME. This nanoprobe-based MRI could be used to monitor hypoxia and further predict the response to radiotherapy and immunotherapy in PDAC [145]. Additionally, Luo et al. developed metabolizable dextran–indocyanine green nanoprobes, which could obtain the dynamic image of deep-seated TAMs in PDAC. Since TAMs in the TME are significantly related to treatment outcomes and prognosis of PDAC, these nanoprobes present great potential for precision therapy, including immunotherapy in PDAC [146]. Furthermore, the immunotherapy process can exhibit specific imaging manifestations, such as immunotherapy hyperprogression [147]. Currently, there are no studies, to our knowledge, reporting on the role of nanoparticles in immunotherapy hyperprogression imaging, which is well worth studying, based on the sensitivity of nanoparticles displayed in the imaging.

In summary, although only a few studies on nanoparticles in imaging and therapeutic aspects of PDAC immunotherapy have been conducted, based on their potential role in these processes, further studies are necessary and valuable.

## 4. Conclusions and Perspectives

As a malignant tumor with an increasing mortality, there is an urgent need to find a more effective therapeutic strategy to improve prognosis for patients with PDAC. Despite the limited clinical efficacy of the current immunotherapy regimens for PDAC, a variety of clinical and preclinical evidence suggests that immunotherapy still holds great promise in the treatment of PDAC. Recently, nanomaterials have shown potential in overcoming current limitations in PDAC immunotherapy. The advantages of nanoparticles include controllable sizes and drug release, prolonged circulation, and co-delivery of multiple therapeutic agents in a site-specific manner. Thus, small-size nanoparticles can penetrate the dense stroma of PDAC to reach deeper parts of the tumor and activate the PDAC response to immunotherapy by regulating the TIME. Additionally, nanoparticles can be used as carriers for several types of immunomodulators, such as microRNAs and siRNAs. Furthermore, some new nanoparticle-based therapies, such as the nanoparticle-based PTT, PDT, CDT, and SDT, have shown promising efficacies in provoking an immune activation and inducing antitumor effects in PDAC.

However, while we recognize the hopes of immunotherapy in PDAC, we should also consider the challenges that nanomedicine may face in this. These challenges include unknown toxicity, selection of the right particle carrier, and translating results from preclinical studies to a clinical setting. Solving these problems will help nanomedicine improve the efficacy of immunotherapy and OS of patients with PDAC. This review provides new research directions for the clinical translation. We expect that nanomedicine will soon cooperate with immunotherapy to improve prognosis of patients with PDAC.

## Figures and Tables

**Figure 1 pharmaceutics-14-02033-f001:**
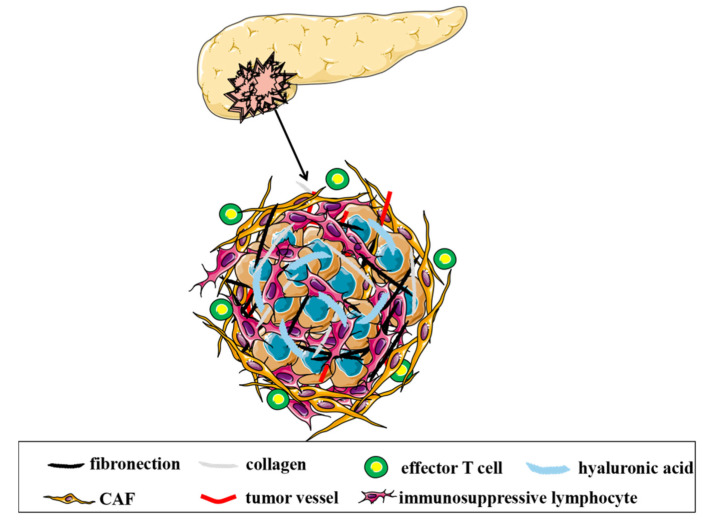
The tumor immune microenvironment (TIME) of PDAC.

**Table 2 pharmaceutics-14-02033-t002:** Nanoparticle-based therapeutic strategies for enhanced PDAC immunotherapy.

Strategy	Agent/Drug	Nanoparticle	Administration	Immune Effect	Publication Year	Ref.
**Nano-based ICD**	oxaliplatin	amphiphilic diblock copolymer nanoparticles	iv	DAMPs↑; DCs↑; CD3+CD8+ cytotoxic T lymphocytes↑	2016	[79]
	oxaliplatin	silicasome nanocarrier	iv	CD8+/FoxP3+ T cell ratios↑;	2021	[80]
	ingenol-3-mebutate (I3A)	2-(3-((S)-5-amino-1-car_x0002_boxypentyl)-ureido) pentanedioate (ACUPA− ) and triphenylphosphonium (TPP+) modified nanomicelles	iv	DCs↑; CD8+ T cells↑; CD4+ T cells↑	2021	[81]
**Nano-based immune modulation**	M2 peptides + miR-125b	hyaluronic acid-poly(ethylene imine) nanoparticles	ip	M1-to-M2 macrophage ratio↑;	2021	[82]
	PI3K-γ inhibitor NVP-BEZ 235 + CSF-1R-siRNA	nanomicelle (M2 TAMs targeting nanomicelles)	iv	M2-TAMs↓; MDSCs↓; M1-TAMs↑; CD8+ and CD4+ T cell infiltration↑;	2020	[83]
	monoacylglycerol lipase siRNA + endocannabinoid receptor-2 (key receptor regulating macrophage phenotype) siRNA	reduction-responsive poly (disulfide amide) (PDSA)-based nanoplatform	iv	repolarization of TAMs into tumor-inhibiting M1-like phenotype	2021	[84]
	paclitaxel-loaded 3-aminophenylboronic acid_x005f_x0002_modified low molecular weight heparin–D-α-tocopheryl succinate	micellar nanoparticle	iv	infiltration of CD4+ T cells and CD8+ T cells↑; MDSCs ↓	2021	[85]
	cyclic diguanylate monophosphate (cdGMP), an agonist of the stimulator of interferon genes (STING) pathway + monophosphoryl lipid A (MPLA), a Toll-like receptor 4 (TLR4) pathway agonist	immuno-nanoparticles	iv	CD45+ immune cells↑; DC cells↑; macrophages↑	2021	[86]
	IL-10 trap + CXCL12 trap	liposome-protamine-DNA	ip	M2 macrophages↓; MDSCs↓; PD-L1+ cells↓; Immuno suppressive plasma cells (ISPCs)↓; activated DCs↑; NK cells↑; CD8+ T cells in tumor↑	2018	[87]
	CXCL12 trap + PD-L1 trap	liposome-protamine-DNA	iv	MDSCs↓; Treg cells↓; accumulated macrophages↓; M1/M2 ↑; T cell infiltration↑	2017	[88]
	Bcl2 double-stranded RNA (dsRNA) + Retinoic acid-inducible gene I (RIG-I)-like receptors	lipid calcium phosphate nanoparticles	iv	Th1 cytokines↑; CD8+ T cells/Treg cells↑; M1/M2 ↑; immunosuppressive B regulatory cells↓	2019	[89]
	IL-12+SBRT	microspheres	intratumoral injection	upregulation of Th1 and antitumor factors IL-12, IFN-γ, CXCL10, and granzyme B	2020	[90]
	PD-L1	poly(lactic-co-glycolic acid;PLGA)-based siRNA nanoparticle	iv	IFN-gamma positive CD8+ T cells↑; Granzyme B+ cell↑;	2021	[91]
	oxygen	microcapsules	intratumoral injection by ultrasound-guided percutaneous injection	oxygen microcapsules + anti-PD-1 antibody: the infiltration of TAMs and polarize pro-tumor M2↓; macrophages into anti-tumor M1 macrophages; the proportion of Th1 cells↑; CTLs↑	2022	[62]
	DCs vaccine	aluminum hydroxide nanoparticle with polyethyleneimine (PEI) modification (LV@HPA/PEI).	subcutaneously vaccinated	CD3+ CD8+ IFNγ+ T cells↑;	2018	[92]
**Nano-based chemo-immunotherapy**	IDO inhibitor, indoximod (IND)+ oxaliplatin (OX)	mesoporous silica nanoparticles(MSNP)	iv	CD8+/Tregs↑; DC maturation↑; cytotoxic T lymphocytes↑; Foxp3+ T cells(Tregs)↓;	2017	[93]
	IDO1 siRNA+oxaliplatin	cationic lipid-assisted nanoparticles	iv	DC maturation↑; tumor-infiltrating T lymphocytes↓; Treg↓; central memory T cells (TCM)↑; effector memory T cells (TEM)↑;	2019	[94]
	GEM + NLG919 (IDO1 inhibitor) + paclitaxel	micelles	iv	CD4+ IFNγ+ T cells↑; CD8+ IFNγ+ T cells↑; Treg cells↓	2020	[95]
	gemcitabine	pH-sensitive polymer	ip and iv	down-regulated the infiltration of macrophages in the tumor tissue; up-regulated the PD-L1 expression on the surface of cancer cells; MDSC↓; CD3+T cells↑; CD8+ T cells↑; CD4+ T cells↑; TAMs↓; Tregs↓;	2021	[96]
	GEM + signal transducer and activator of transcription 3 (STAT3) inhibitor (HJC0152)	micelle	iv	reversing M2 to M1-TAMs, M1/M2↑; downregulating MDSCs/Tregs, and upregulating Teff	2022	[97]
	celastrol (CLT) + 1-methyltryptophan (MT, IDO inhibitor)	hyaluronic acid coated cationic albumin nanoparticle	iv	CD4+ T cells in the spleen↑; CD8+ T cellsin the spleen↑;	2019	[98]
	irinotecan+Anti-PD-1 antibody	silicasome	iv(irinotecan) + ip(anti-PD-1)	irinotecan by the silicasome: CD8+ T cells/Treg cells↑; PD-L1 expression↑;	2021	[99]
	galectin-9 siRNA + oxaliplatin	exosome	iv	M2-TAMs↓; Tregs↓; M1-TAMs↑; cytotoxic T cells and helper T cells↑; CD8+ CTLs↑; mature DC ↑;	2020	[22]
	gemcitabine-loaded poly (lactic-co-glycolic acid) (PLGA) nanoparticles + M2-like macrophages peptides (M2pep) +PD-L1 antibody	biomimetic dual-targeting nanomedicine	iv	elimination of PD-L1-positive macrophages and the downregulation of PD-L1 expression; reprogrammed macrophages, downregulated PD-L1 expression, and sustained T cell populations,	2022	[100]
Nano-based stoma modulation	α-mangostin + LIGHT (tumor necrosis factor superfamily 14)	calcium phosphate liposome	iv	CD45+CD3+CD8+ T cells↑; CD45+CD3+CD4+ T cells ↑; CD45+B220+ B cells↑; Tregs and F4/80+ macrophages↓; CD8+ T/CD4+ T↑; CD4+T/Treg ↑; induces the intratumoral tertiary lymphoid structures	2020	[101]
	TGF-β receptor inhibitors (LY2157299) + siRNA targeting PD-L1 (siPD-L1)	acidic tumor extracellular pH (pHe) responsive clustered nanoparticle	iv	tumor infiltrating CD8+ T cells↑; IFN-γ↑;	2020	[102]
	sonic hedgehog inhibitor (cyclopamine) + cytotoxic chemotherapy drug (paclitaxel)	polymeric micelles	iv	infiltration CD8+T cells↑;	2018	[103]
	PEGPH20+GVAX	PEGylated recombinant human PH20 hyaluronidase	iv	infiltrating CD3+CD8+ T cells↑; infiltrating CD3+CD4+ T cells↑	2019	[104]
	PEGPH20+FAKinhibitor+anti-PD-1 antibody	PEGylated recombinant human PH20 hyaluronidase	iv + oral gavage + ip	Increases T cell infiltration and alters T cell phenotype towards effector memory T cells.	2022	[105]
	paclitaxel + phosphorylated gemcitabine	codelivery micelles	iv	cytotoxic T cells↑; T helper cells↑; Tregs cells↓; tumor infiltration by the CD4+ and CD8+ T cells↑;	2019	[106]
	CXCR4 antagonist + anti-miR-210 + siKRASG12D	cholesterol-modified polymeric	ip	CD8+ T cells infiltration↑; M2 TAMs↓;	2020	[107]
Nano-based PTT/PDT/CDT/SDT	mild hyperthermia + immune checkpoint blockade (BMS-202)	size-adjustable thermo-and fibrotic matrix-sensitive liposomes	iv + laser irradiation	DC maturation↑; infiltration of CD4+ T cells↑; infiltration of CD8+ T cells↑; Tregs↓; chemokines (IL-6, IFN-γ) ↑	2021	[70]
	immunomodulatory thymopentin + near-infrared indocyanine green	self-assembly nanoparticle	in situ injection + laser irradiation	CD3+CD4+ T cells↑; CD3+CD8+ T cells↑; CD4+IL-4+ T cells ↑; CD8+INF-γ+ T cells↑;	2021	[71]
	GEM+NLG919 (IDO inhibitor)	polydopamine (dp)-coated nanoparticles	iv + laser irradiation	intratumoral infiltration of CD8+ T↑; (IFN-γ) in CTLs↑; GZB positive NK cells↑	2021	[72]
	indocyanine green (ICG,photothermal agent) + imiquimod (IMQ,toll-like-receptor-7 agonist)	iron oxide nanoplatform	iv + laser irradiation	polarization of macrophages to the M1 phenotype; tumor infiltration of T cells↑; Tregs↓; CD8+ T cells/Treg cells↑; CD4+ T cells/Treg cells↑	2022	[73]
	bromodomain-containing protein 4 inhibitor (BRD4i) JQ1+ cyclodextrin-grafted hyaluronic acid (HA-CD) + pyropheophorbide a (PPa)	supramolecular prodrug nanoplatform	iv + laser irradiation	DC maturation↑; intratumoral infiltration of CD8+ T ↑; intratumoral infiltration of Tregs ↓ memory T lymphocytes (TEM) infiltration↑	2021	[74]
	chlorin e6 photosensitizer	tumor-derived exosomes	iv + laser irradiation	release of cytokines from immune cells	2021	[75]
	Rose Bengal+ CaO_2_ nanoparticle	pH-sensitive polymethacrylate-coated CaO_2_ nanoparticle	iv + ultrasound	CD8+T cells↑; Treg↓	2021	[76]
	cavitation-assisted endoplasmic reticulum targeted sonodynamic droplets+PD-L1 antibody	nanodroplets	iv + ultrasound	DC maturation↑	2022	[77]
	4-(phosphonooxy)phenyl-2,4-dinitrobenzenesulfonate + Fe3 + hyaluronic acid decoration	chemodynamic nanocomplex	iv	induced the polarization of the M2 phenotype back to the M1 phenotype	2020	[78]

iv: intravenous administration; ip: intraperitoneal administration; ↑: increase in number; ↓: decrease in number.

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
