# Peer review of "Nanoparticle-Based Therapeutic Strategies for Enhanced Pancreatic Ductal Adenocarcinoma Immunotherapy"

_pharmaceutics, 2022, doi:10.3390/pharmaceutics14102033_

Round 1

Reviewer 1 Report

It is an excellent review of nanoparticle-based strategies to overcome resistance of PDAC. However, one very important aspect was touch very little, only. Imaging and theranostics is an excellent opportunity for nanoparticles to be a more precise medicine, therefore I would suggest to authors to expand a little bit on that.

Incorporation of Copper-64 allows to visualize nanoparticles by PET (PMID: 36014656).

Radiolabeling of nanoparticles with Lutetium-177 allows for using them as theranostic agent (PMID: 35889715).

There are also gold nanoparticles, which were used for imaging, but could also be decorated with anti-cancer agents (PMID: 28713230).

There are also so many examples of benefits that nanoparticles can be relatively easily visualized using various contrast strategies, which is very difficult to small molecules and even to macromolecules.

Author Response

Thank you very much for your positive comment and kind suggestion. In response to this suggestion, we reviewed the literature and found several reports on the role of nanoparticles in PDAC imaging, which were also thematic in previous reviews(PMID: 30445002; PMID: 33854877). Combined with the topic of our review, we added the content of nanoparticles in imaging and theranostic of PDAC immunotherapy (subsection 3.5). We hope the revised manuscript version could be acceptable for you.

Reviewer 2 Report

The manuscript is a comprehensive review of PDAC both current and potential future therapies are described. PDAC and the treatment problems are extensively described addressing the immune exclusion due to the dense stroma and the hypoxic and drug-inaccessible environment. 

The  various nanoparticle based therapies were also extensively and well described. However, I was missing a section on how nanoparticle based therapeutics can access the tumour. It is well known that drugs, antibodies and cells are excluded from the dense TME in PDAC, so  how can nanoparticles with drugs access the tumour supposedly penetrating the stroma? 

The authors need to clearly explain this in an additional section/paragraph and also edit the language (minor changes) prior to acceptance for publication. 

Author Response

Thank you very much for your positive comments and kindly criticism. It was our oversight that did not point out separately the role of nanoparticles in penetrating the dense stroma of PDAC. Therefore, in this revision, we added this section in the first paragraph of the third part of the article. We also marked the role of nanoparticles in the stroma modulation of PDAC in various literature in section 3.4. At the same time, we invited an expert from native-speaking countries to revise the English of this article. We hope this revision can satisfy you.

Reviewer 3 Report

The authors have done a fantastic job on summarizing the various nanoparticle-based treatment modalities tested to enhance the efficacy of immunotherapy in PDAC. It is well written and easy to follow. Including a list of acronyms and their expansions would be helpful to the readers. Also, check for errors, the review needs minor edits.

Author Response

Thank you for your positive comments on our article. We have added a list of abbreviations and re-examined the article according to your suggestion, including the language part. We hope our revision can satisfy you.